# Trends in Health Communication: Social Media Needs and Quality of Life among Older Adults in Malaysia

**DOI:** 10.3390/healthcare11101455

**Published:** 2023-05-17

**Authors:** Hana W. Jun Chen, Roy Rillera Marzo, Nur Hafizah Sapa, Absar Ahmad, Haryati Anuar, Mohammed Faez Baobaid, Nurul Akmal Jamaludin, Hazian Hamzah, Siamak Sarrafan, Hassan Omar Ads, Kavitha Ashok Kumar, Jalal Hadi, Hafsah Sazali, Mohammed A. Abdalqader

**Affiliations:** 1International Medical School, Management & Science University, Shah Alam 40100, Selangor, Malaysia; 2Global Public Health, Jeffrey Cheah School of Medicine and Health Sciences, Monash University Malaysia, Petaling Jaya 47500, Selangor, Malaysia; 3School of Graduate Studies, Management & Science University, Shah Alam 40100, Selangor, Malaysia; 4College of Veterinary Science and Animal Husbandry, Birsa Agricultural University, Ranchi 834006, Jharkhand, India; 5School of Medical Sciences, Universiti Sains Malaysia, Kota Bharu 16150, Kelantan, Malaysia; 6Faculty of Medicine, University of Cyberjaya, Cyberjaya 63000, Selangor, Malaysia

**Keywords:** social media, quality of life, elderly, older adults, health communication

## Abstract

Background: While social media continues to dominate, social media platforms have become powerful health communication tools for older users. However, fulfilling their social media needs can be both detrimental and beneficial to their quality of life (QoL). This study assessed social media needs as they relate to QoL among older adults in Malaysia. Methods: We conducted a cross-sectional study and adopted convenience sampling to recruit participants. The participants were required to self-report their sociodemographic profile, social media use and needs, and QoL. Social media use and needs were assessed using the Social Networking Sites Uses and Needs (SNSUN) scale, and QoL was assessed using the WHOQOL-BREF questionnaire. Multiple linear regression was performed to identify the predictors of QoL. Results: The findings revealed that the fulfilment of social integrative needs was the strongest predictor of higher QoL in all domains. However, those using social media for their affective needs demonstrated lower psychological health quality. Conclusions: Fulfilling social integrative needs is the key to improving the QoL among older adults. The continuous development of age-friendly applications is essential to keep up with constantly changing social media trends and bridge the gap of social media inequalities. More importantly, it would enable older adults to utilize social media to its fullest potential and enjoy a higher QoL through accessible health communication tools.

## 1. Introduction

With the advent of the Internet, the utilization of social media has become almost inevitable. Social media refers to Internet-based platforms that allow users to interactively share their thoughts, experiences, and information [1]. The immense growth of the Internet over the past decade has been most evident in Asia [2]. In 2023, global data showed that more than half of the world’s population (4.76 billion) were social media users [2]. In the digital age, the digital divide is a major concern as it leads to unequal access to digital technology, including the Internet and social media, especially among older adults [3,4]. According to the Malaysian government, older adults (or senior citizens) are defined as individuals who are aged 60 years and older, a definition in line with that of the 1982 World Assembly on Ageing in Vienna [5]. Unfortunately, the world’s population is aging, and the situation is no different in Malaysia. In 2020, the Department of Statistics Malaysia estimated that 10.7% (3.5 million) of the Malaysian population were aged 60 and above [6]. According to the population census, Malaysia’s aging population is expected to double from 7% in 2005 to 14% in 2028 [7]. As the number of older adults in Malaysia continues to rise, our research aims to explore different types of social media needs and their potential roles in improving quality of life (QoL) among older adults in Malaysia, thereby highlighting factors that can bridge the digital divide.

In Malaysia, social networking is the second most common online activity, engaging 93.3% of all Malaysian Internet users [4]. According to the Internet Users Survey 2020 by the Malaysian Communications and Multimedia Commission (MCMC), 11.3% of Malaysians were at the time non-Internet users, of which 51.8% were aged 60 and above [4]. The primary reasons given for not using the Internet were lack of interest, being too old to learn, and not having a device [4]. In 2022, there were 30.2 million (91.7%) social media users in Malaysia [8]. Although there is no actual data on social media usage among those aged 60 and above in Malaysia, Statista’s data from 2020 revealed that only 3.4% of this age group were Internet users [9]. The MCMC also reported a decline in Internet usage among Malaysian adults aged above 45 years from 2018 to 2020 [4]. Given the current situation, older adults will continue to be a vulnerable group if the digital divide persists.

The term “use” refers to the act of using something for a specific purpose, while “needs” are the essential things that a person must have to live a satisfactory life [10]. Therefore, this study examines the interconnection between “the use of social media and the satisfaction derived from social media use” and QoL. The World Health Organization (WHO) defines QoL as an individual’s perception of their position in life within the context of the culture and values systems in which they live, as well as their goals, expectations, standards, and concerns [11]. A recent review has suggested that older adults who use the Internet have higher levels of life satisfaction, well-being, and QoL [12]. Social media use allows users to meet various needs, such as diversionary, cognitive, affective, personal integrative, and social integrative needs [13]. The fulfilment of social media needs, such as obtaining information, self-expression, and entertainment [14], plays a vital role in reducing loneliness and enhancing the QoL of adults aged 50 years and older [15]. Furthermore, the evolving role that social media played in health promotion during the COVID-19 pandemic lockdown further highlights the importance of social media as a health communication platform for improving public health and QoL [16,17].

Currently, the WHO focuses on four domains of QoL: physical health, psychological health, social relationships, and environmental health [18]. Although the literature on social media needs and QoL among older adults is generally scarce, numerous studies have linked the fulfilment of social media needs with many advantages that are closely related to the different domains of QoL [14,19]. For physical health, social media enables older adults to access information such as diet plans, pain management techniques, and exercise routines, and these can help reduce morbidity of chronic diseases [14]. Social media also contributes to health maintenance, interpersonal relationships, independence, and cognitive functioning, all of which are essential for quality living [19]. Through social media, older people can communicate with healthcare professionals remotely and seek information to improve their general well-being, mental health, and QoL [20,21].

The use of social media allows older adults to have better social relationships and mental health through connections with family, friends, or a large social network, increased access to information, and engagement in online leisure activities [12,21]. Social networking sites also provide social support that serves as an effective buffer against stress through supportive social networks [22]. Given this backdrop, knowledge regarding the relationship between social media use and needs and QoL is still lacking. Therefore, this study assessed the predictors of QoL among older adults in Malaysia. Specifically, we explored (1) whether demographic and socioeconomic predictors and (2) specific needs fulfilled through social media use affect the QoL of older adults in Malaysia. This will shed light on how best to re-strategize an inclusive health communication platform using social media across the stages of old age.

## 2. Materials and Methods

### 2.1. Participants

The participants comprised Malaysians aged 60 years and above who resided in Malaysia and included individuals with different levels of income (low, middle, high), education (primary, secondary, post-secondary, tertiary), and employment (employed, unemployed, retired). Convenience sampling was used to enroll potential participants. The sample size was calculated using a single population proportion calculator—Epitools (https://epitools.ausvet.com.au/oneproportion (accessed on 16 January 2022)). Based on a 5% desired precision and a 95% confidence level, and assuming that 51.8% of non-Internet users were aged 60 years and above [4] and that 3.5 million of the Malaysian population were aged 60 years and above [6], a sample size of 384 older adults was required. An additional possible non-response rate of 20% was also accounted for, necessitating a minimum sample size of 461. The study included participants that were literate in English or Malay, had access to the Internet during the data collection period, and were free from clinically diagnosed mental illnesses. Older adults living in old folks’ homes, nursing homes, or senior care centers were excluded from the study.

### 2.2. Design and Procedure

This cross-sectional study was conducted from March to May 2022 in Malaysia using a quantitative-based approach. An anonymous, validated, self-administered online questionnaire was distributed using the remote data collection (RDC) method. The online questionnaire was hosted on Google Forms and disseminated through social media platforms. The online questionnaire was shared with non-Internet or non-social media users via their family members or peers. Participants were asked to self-report their sociodemographic profile, social media use and needs, and QoL. The questionnaire was piloted on a sample of 30 to test its validity and reliability, and the data obtained from the pilot study were not included in the final analysis. The online survey allowed only one response per participant. Data encryption was used to ensure data confidentiality.

### 2.3. Study Variables and Tools

This study utilized a validated questionnaire consisting of 56 items and was divided into 3 sections: Section A—demographic and socioeconomic profile (10 items); Section B—social media use and needs (SNSUN scale) (adapted from Ali et al., 2019 [13], 20 items); and Section C—QoL (WHOQOL-BREF) (adapted from WHOQOL Group, 1998, 26 items) [18]. The online survey fulfilled the criteria of the Checklist for Reporting Results of Internet E Surveys (CHERRIES).

Both Malay and English versions of the questionnaire were available. The WHOQOL-BREF has already produced a validated Malay version [23]. The back-to-back translation of the SNSUN scale was carried out by three independent translators who are native Malay speakers with expertise in our research area. Initially, the original SNSUN questionnaire was translated into Malay by one translator and then translated back into English by an independent translator who was blinded to the original questionnaire. Then, another expert compared the two English versions and discussed any discrepancies to reach a consensus on the final translated questionnaire. To ensure that the translated SNSUN questionnaire was culturally relevant to our target population, the translated SNSUN questionnaire was reviewed by a focus group of local experts in our research area for cultural appropriateness, and feedback was provided. The authors incorporated their suggestions and further refined the translated SNSUN questionnaire.


**Demographic and socioeconomic profile**


Ten demographic and socioeconomic factors were collected for this study: age, gender, marital status, highest qualification level, household income group, income source, employment status, living arrangement, residential area, and health condition.


**Social Media Use and Needs**


The Social Networking Sites Uses and Needs (SNSUN) scale is a self-administered instrument used to assess social media use and needs [13]. This study adapted twenty items from the questionnaire: 1 item from social media use and 19 items from social media needs, including diversions (3 items), cognitive needs (4 items), affective needs (3 items), personal integrative needs (4 items), and social integrative needs (5 items). The item for social media use has four possible responses: “yes”, “occasionally”, “rarely”, and “no”. As for social media needs, each question is graded on a four-point Likert scale where 1 represents “strongly disagree” and 4 represents “strongly agree”. Meaningful classifications were developed for social media needs, and the following cut-off scores were obtained: diversion (≤6 = low diversion; >6 = high diversion); cognitive needs (≤8 = low cognitive needs; >8 = high cognitive needs); affective needs (≤6 = low affective needs; >6 = high affective needs); personal integrative needs (≤8 = low personal integrative needs; >8 = high personal integrative needs), and social integrative needs (≤10 = low social integrative needs; >10 = high social integrative needs).

The SNSUN scale has been psychometrically tested and has shown good reliability and validity for SNS needs [13]. Expert validation was performed, and the content validity index (CVI) values for relevancy and clarity for SNS needs were within an acceptable range (between 0.71 and 1.00). Eighteen items from the five dimensions of SNS needs (diversionary, cognitive, affective, personal integration, and social integration) showed high reliability with Cronbach’s alpha values of 0.922 [13].


**Quality of Life**


The WHOQOL-BREF questionnaire is a self-administered tool comprising 26 items that measure QoL [18]. The WHOQOL-BREF is a shorter version of the WHOQOL-100, which the WHO developed and published in 1995. The 26 items of the questionnaire assess an individual’s perceptions of their health and well-being over the previous two weeks. Responses to questions are provided using a five-point Likert scale where 1 represents “disagree” or “not at all” and 5 represents “completely agree” or “extremely”. Among the 26 items, 2 separate questions ask specifically about (1) overall perception of health and (2) overall perception of QoL, and the other 24 questions cover the 4 domains of QoL: physical health (7 items), psychological health (6 items), social relationships (3 items), and environmental health (8 items). Interpretation of the scoring systems is as follows: overall QoL ranges from 1 “very poor” to 5 “very good”; general health ranges from 1 “very dissatisfied” to 5 “very satisfied”; and for physical health, psychological health, social relationships, and environmental health, 0–25 represents “poor”, 26–50 represents “medium”, 51–75 represents “good”, and 76–100 represents “excellent”.

The WHOQOL-BREF has shown good discriminant validity, content validity, internal consistency, and test–retest reliability [18,24,25]. The WHOQOL-BREF has demonstrated significant differences in discriminating between ill and well subjects in all domains. Concerning reliability, the Cronbach’s alpha values for each of the four domain scores ranged from 0.66 to 0.84, indicating good internal consistency. The test–retest reliabilities for the four domains were 0.66 for physical health, 0.72 for psychological health, 0.76 for social relationships, and 0.87 for environmental health.

The Malaysian adaptation of the WHOQOL-BREF has also been psychometrically tested. The Malay version showed good reliability and construct validity, with factor analysis supporting the four-domain structure of the questionnaire [23]. The Cronbach’s alpha values for the four domains showed good internal consistency: physical health: *α* = 0.65; psychological health: *α* = 0.63; social relationships: *α* = 0.68; and environment: *α* = 0.80.

For the current study, both the SNSUN and WHOQOL-BREF scales demonstrated good content validity, face validity, and reliability. A panel of 3 experts evaluated the content validity of the questionnaires (20 items from the SNSUN scale and 26 items from the WHOQOL-BREF scale). All the questions received an acceptable CVI of over 70%, resulting in a final CVI of 100%. Face validity and reliability were assessed through a pilot study involving 30 subjects. The final FVI was 89%, and the internal consistency of all sections was good, with Cronbach’s alpha values ranging from 0.61 to 0.90 for SNSUN (diversion: *α* = 0.87; cognitive needs: *α* = 0.80; affective needs: *α* = 0.61; personal integrative needs: *α* = 0.90; and social integrative needs: *α* = 0.75) and Cronbach’s alpha values ranging from 0.75 to 0.88 for WHOQOL-BREF (physical health: *α* = 0.75; psychological health: *α* = 0.78; social relationships: *α* = 0.88; and environment: *α* = 0.82).

### 2.4. Data Analysis

Descriptive statistics were used to describe the demographic and socioeconomic data, social media use, needs, and QoL. An independent *t*-test was used to identify the association between social media needs and QoL. Multiple linear regression analysis was performed to identify the predictors of QoL. The outcome variables (the four domains of QoL) and predictor variables (social media needs) were compared after controlling the demographic and socioeconomic factors.

The multiple linear regression equation is as follows:*Y* = *β*_0_ + *β*_1_*X*_1_ + *β*_2_*X*_2_ + *β*_3_*X*_3_ + … + *β*_10_*X*_10_ + *ε*
where *Y* is the QoL, *β*_0_ is the intercept value, *X_i_* are the ten independent variables (age, gender, marital status, highest qualification, household income group, income source, employment status, living arrangement, residential area, and health condition), and *β_i_* are the estimated regression coefficients of the respective independent variables. *ε* is the model error, i.e., the variation between our estimate of *Y* and the actual value.

Statistical analyses were performed using Statistical Package for Social Sciences (SPSS) statistical software version 26.0, and a value of *p* < 0.05 was considered statistically significant.

### 2.5. Ethical Considerations

This study obtained ethical approval from the Ethics Committee of Management and Science University (Ethics Code: MSU-RMC-02/FR01/11/L1/109). Informed consent was obtained from all the respondents before distributing the survey questions.

## 3. Results

### 3.1. Demographic and Socioeconomic Profile

A total of 826 out of 907 (91.1%) participants completed the survey. The characteristics of the study sample (*n* = 826) are shown in Table 1. The mean age of the participants was 67.6 (6.4) years, ranging from 60 to 96 years. More than half were female (51.6%), and the majority were married (69.2%), currently lived with their families (86.8%), and resided in urban areas (70.3%). Most of the participants had a secondary level of education (35.2%) and belonged to the low-income group (64.2%). About two fifths were retired (42.9%), and the main source of income was mostly from children (34.7%). More than half of the participants were living with chronic illnesses (54.1%).

### 3.2. Social Media Use and Needs

Table 2 presents the social media needs of the participants. Of the 826 participants, 638 (77.2%) were using social media and reported having social media needs. The respective mean scores and SDs of the social media needs were 8.42 and 1.88 for the diversionary needs, 11.92 and 2.37 for the cognitive needs, 7.06 and 2.00 for the affective needs, 9.00 and 2.73 for the personal integrative needs, and 15.53 and 2.74 for the social integrative needs). Among all the different social media needs, the highest needs for the participants were social integrative (95.5%) and cognitive needs (87.1%).

### 3.3. Quality of Life

Table 3 shows the QoL of the participants. The respective mean scores and SDs of the four self-reported QoL domains were 62.55 and 15.15 for physical health, 66.99 and 13.33 for psychological health, 67.09 and 18.13 for social relationships, and 68.92 and 13.40 for environmental health. Almost one fourth of the participants self-reported having a poor QoL (24.6%), and as many as 40.3% reported poor health satisfaction (a score below 4 is considered poor for QoL and health satisfaction).

### 3.4. Association between Social Media Needs and Quality of Life

Table 4 presents the differences in mean scores across the four QoL domains with social media needs. All five of the social media needs categories were significantly associated with the physical health domain of QoL: diversionary needs: *t* = −2.493, *p* = 0.013; cognitive needs: *t* = −3.150, *p* = 0.002; affective needs: *t* = −2.419, *p* = 0.016; personal integrative needs: *t* = −2.631, *p* = 0.009; and social integrative needs: *t* = −3.056, *p* = 0.002. Social media needs for cognitive (*t* = −2.330, *p* = 0.020) and social integrative needs (*t* = −2.889, *p* = 0.004) were significantly associated with the psychological health domain of QoL. A significant relationship with the environmental health domain of QoL was only observed in one social media needs category (social integrative needs: *t* = −2.459, *p* = 0.014).

### 3.5. Predictors of Quality of Life

A multiple linear regression analysis was performed to identify the predictors of QoL. The outcome variables (the four domains of QoL) and predictor variables (social media needs) were compared after controlling the demographic and socioeconomic factors (Table 5). A multiple linear regression model based on ten determinants explained 22% of the variation in physical health quality (*R*^2^ = 0.223, adjusted *R*^2^ = 0.195, *F* = 8.006, *p* < 0.001), 14% of the variation in psychological health quality (*R*^2^ = 0.139, adjusted *R*^2^ = 0.108, *F* = 4.520, *p* < 0.001), 8% of the variation in social relationship quality (*R*^2^ = 0.080, adjusted *R*^2^ = 0.047, *F* = 2.440, *p* < 0.001), and 7% of the variation in environmental health quality (*R*^2^ = 0.068, adjusted *R*^2^ = 0.034, *F* = 2.028, *p* < 0.01). For each model, according to the four domains of QoL, the null hypothesis of the linear regression’s *F*-test states that the model explains zero variance in the dependent variable. The *F*-test is highly significant, which means the model explains a significant amount of the variance in the respective QoL domain.


**Demographic and socioeconomic predictors**


Living without chronic illness is the strongest predictor for all four QoL domains and is generally associated with better physical, psychological, and environmental health quality as well as better social relationships. Males (*β* = −0.086, 95% CI: −4.39; −0.129) were less likely to have better psychological health quality than females. Married individuals (*β* = 0.180, 95% CI: 0.388; 14.819) were more likely to report better social relationship quality, while divorced individuals (*β* = −0.195, 95% CI: −12.554; −1.44) were less likely to have better physical health quality than single people. Individuals who had completed at least a secondary education (*β* = −0.145, 95% CI: −7.153; −0.815) were less likely to have better psychological health quality, and those with tertiary degrees were less likely to have better QoL in three out of the four QoL domains (psychological health, social relationships, and environmental health). People who received their income from their children (*β* = 0.136, 95% CI: 1.276; 7.456) were more likely to have better physical health quality, while those with businesses or investments (*β* = 0.087, 95% CI: 0.223; 5.499) were more likely to have better psychological health quality than those who relied on their job as an income source. Employed individuals were more likely to have better psychological health (*β* = 0.120, 95% CI: 0.249; 6.812) and social relationship quality (*β* = 0.133, 95% CI: 0.742; 10.181) than unemployed individuals. No associations were observed between QoL and age, household income, living arrangement, and residential area.


**Social media needs predictors**


Even after controlling for demographic and socioeconomic predictors, social media needs were associated with QoL. Social integrative needs were significantly associated with all the QoL domains and were the strongest predictor for a higher QoL in all the domains. Older adults using social media to stay connected with family and friends (higher social integrative needs) demonstrated higher QoL. Individuals using social media to acquire information and knowledge (higher cognitive needs) were more likely to have higher physical health quality (*β* = 0.109, 95% CI: 0.116; 1.208), while individuals using social media for emotional support or dating (higher affective needs) demonstrated lower psychological health quality (*β* = −0.168, 95% CI: −1.786; −0.414).

## 4. Discussion

Due to the global trend of rapidly aging populations, there is an increasing need for effective health communication tools to improve the health care, daily life support, and QoL of older adults [20,21]. To the best of our knowledge, this is the first study conducted in Malaysia which explores social media needs and their relationships with QoL among older adults, addressing the digital divide of this vulnerable group. The mean age of the participants was 67.6 (6.4) years (youngest-old) [27], which helps explain why over half (77.2%) of the older adults in Malaysia use social media and report having social media needs. The highest social media needs were social integrative needs (95.5%) and cognitive needs (90.9%). Living without chronic illness was associated with better QoL in all domains. The study found that, aside from sociodemographic factors, social media needs were associated with QoL. The fulfilment of social integrative needs was the strongest predictor of a higher QoL in all domains. Using social media to stay connected with family and friends (higher social integrative needs) was associated with a higher QoL in all domains. Conversely, using social media for emotional support or dating (higher affective needs) had a detrimental impact on the psychological health quality of older adults.

The sociodemographic predictors of QoL revealed that males were less likely to have better psychological health quality than females. Men’s health has been recognized as a significant health issue worldwide since the late twentieth century [24]. Aside from physical well-being, psychological and social well-being are important aspects of men’s health [28]. According to research, unhealthy lifestyles and risky behaviors are the main factors behind male mortality from non-communicable diseases (NCDs). Similarly, men prefer to self-medicate and engage in risk-taking behaviors to cope with their problems [29]. Men are more likely than women to keep their problems to themselves rather than seeking professional help [29]. Loneliness and social isolation can ultimately reduce the psychological health quality of older men, which may contribute to their higher mortality rate and lower life expectancy compared with women [30,31].

Our study found that married individuals were more likely to have better quality social relationships than single individuals, an observation which is consistent with the findings of previous studies [32,33]. A study conducted in the Netherlands found that better social relationship quality among married individuals was associated with intimacy and social relationships between partners [33]. The study showed that almost half of individuals aged 50 years and above did not perceive any changes in intimacy over the years [33]. As intimacy is a continuous human need for most individuals, fulfilling marital intimacy also provides better social relationships and social support. The findings of our study are particularly important for older women, who tend to have higher life expectancies than men and may experience reduced social relationship quality when living alone in later years. Thus, a senior citizen-friendly social media platform should be developed to meet the needs of older women and improve their QoL.

Regarding socioeconomic status (SES), our results suggest that individuals with tertiary degrees are less likely to have better psychological health, social relationships, and environmental health quality. This could be due to the demographic profile of the study, with the majority of the participants residing in urban areas. Although our study did not find an association between income and QoL, it is known that better education provides career opportunities with higher salaries, and this is often related to higher levels of occupational stress and poorer workplace relationships [34]. Regardless of job type, our study found that employed individuals had better psychological health and better quality social relationships than unemployed individuals. According to a study conducted in northern Sweden, social determinants of health such as socioeconomic status, economic resources, social networks, and trust in institutional systems are contributing factors to mental health inequalities between unemployed and employed individuals [35]. Unemployment and re-employment during older adulthood can be distressing due to age discrimination in the labor market and difficulties in adapting to new technology [35]. A study conducted in Malaysia during the COVID-19 pandemic found that unemployed individuals had lower digital health literacy and engaged in less health information-seeking behavior [36]. These factors, alongside the social stigma of unemployment, further reduce the psychological health and social relationship quality of this vulnerable group and increase their risk of developing mental health problems.

Aside from employment status, income source is another important factor related to the QoL of older adults. Our study revealed that individuals who receive their income from their children were more likely to have better physical health, while those with businesses or investments typically had better psychological health compared with individuals who relied on their job as an income source. In general, aging is a natural process accompanied by a reduction in body functions [37]. This leads to a gradual decrease in work ability and physical and mental capacity, as well as a growing risk of disease, which subsequently reduces the satisfaction and QoL of older adults [37,38]. Aging also increases physical and financial dependencies as older adults lose their ability to work, lose their jobs, and, most importantly, their primary income sources [37,38]. Consequently, having a secondary income source becomes a necessity to cover daily expenses and the high cost of medical and nursing care during the stage of older adulthood [39]. Nowadays, most older adults with comorbidities need to rely on their children or family caregivers for financial support to cover their medical expenses and nursing care [39]. Some older adults also receive financial aid from different channels, including compensation, insurance claims, investments, and donations from non-profit organizations. Hence, it is understood that older adults with a secondary income source typically enjoy better health than those who rely on their job as their primary income source.

Additionally, this study found that living without chronic illness was associated with better QoL in all domains [40,41]. Aging is one of the major risk factors for many NCDs and chronic inflammatory diseases [37,42]. The majority of older adults suffering from chronic illnesses experienced reduced body function and capacity, giving rise to poorer physical and psychological health [37,38,40,41]. A study conducted in Denmark found that high observed activities of daily living (ADL) motor ability was associated with high QoL among advanced cancer patients [41]. In general, most NCDs are accompanied by a decline in functional status, leading to the loss of the ability to perform ADL [41]. When older adults with chronic illnesses increase their ADL and financial dependency, relying on their family members for physical aid and secondary income, feuds often result [43]. Indeed, disease-specific family conflicts were associated with a lower general QoL [43]. Some children just send their aged parents to old folks’ homes, despite poor environments and caregiver issues [44]. In certain cases, conflicts may involve domestic violence, older adults abuse, and neglect [44]. All of these consequently reduce the QoL of older adults in all four domains (physical health, psychological health, social relationships, and environmental health quality) [44].

This study has demonstrated that, in addition to sociodemographic characteristics, social media needs are associated with QoL among older adults. The fulfilment of social integrative needs was the strongest predictor of a higher QoL in all domains. Social integration is the degree to which an individual is involved in social exchanges with others, whether with family, social networks, or within their communities, and it fosters a sense of belonging [45]. Social integration is a major concern in old age and an important factor for active aging [46]. It has been found to alleviate loneliness, improve physical and mental health, and promote life satisfaction [46,47]. The use of social media to fulfil social integration will help older adults stay connected with their family and friends. With greater social integration, older adults will be able to build larger networks of close relationships, which are essential in shaping health, well-being, and QoL in later life [48,49]. Given the important role of social media in the fulfilment of social integrative needs, we must emphasize the need to close the digital divide for older adults [46]. Through the fulfilment of social integrative needs using social media, older adults will be able to build social integration and increase access to all areas of community life, enabling them to enjoy a better QoL physically, mentally, socially, and environmentally.

Another crucial category of social media needs is cognitive needs. Our study found that older adults who use social media to acquire information and knowledge were more likely to have better physical health. Due to the social isolation and lockdowns that accompanied the COVID-19 pandemic, the adoption of information technology for communication and the dissemination of information has been more important than ever [50,51]. Since the COVID-19 pandemic, the Ministry of Health Malaysia has been promoting health and disseminating health information via social media [52]. This explains why in our study we observed that individuals who have access to social media and know how to use it to fulfil their cognitive needs experience better physical health. Thus, it is understood that older adults who obtain health information from social media platforms generally have better control over their health, resulting in better physical health compared with those who do not use social media to fulfil their cognitive needs. However, with the existing digital gap, digital information alone is insufficient for reaching vulnerable populations such as older adults. Since the emergence and re-emergence of infectious diseases is inevitable, going digital is a necessity now more than ever before [50,53,54]. In a nutshell, the existing digital gap for older adults must be address as social media is the new trend in health communication.

## 5. Limitations and Recommendation for Future Research

Although this study provides important insights, it has some limitations. Firstly, selection bias may have influenced the results, as the sample was predominantly composed of participants from specific demographic backgrounds and settings and may not have been representative of the entire population. To mitigate selection bias in future studies, researchers should aim to recruit a more diverse and representative sample, considering factors such as age, gender, socioeconomic status, and geographical location. Secondly, the validity and reproducibility of the tool used in this study merit further examination. Although the instrument has been used previously in similar research, it is essential to confirm its reliability and validity across different populations and settings. Future research should include the evaluation of the tool’s psychometric properties, such as internal consistency, test–retest reliability, and criterion validity. Additionally, alternative tools or methods could be considered to triangulate findings and ensure their robustness. Lastly, the external validity of the results and conclusions should be thoroughly assessed. The generalizability of the findings to other populations or settings may be limited due to the specific characteristics of the sample and the context in which the study was conducted. However, the robust methodology and large sample size of this study improve the external validity and generalizability of the findings. To enhance the external validity of future research, investigators should conduct similar studies in different populations and settings. Overall, it would be valuable to perform a meta-analysis or systematic review of the existing literature to synthesize the evidence and identify common trends and discrepancies.

## 6. Conclusions

In conclusion, our study reveals that fulfilling social integrative needs via social media is crucial for improving the QoL of older adults in Malaysia. As technology advances and preparations are made for a future pandemic, social media has become an important communication tool for social support, social integration, and finding health information. These findings hold significant implications for policy making and planning and may help to improve older adults’ access to technology and social media in an increasingly digital world. To promote a higher QoL, it is essential to develop age-friendly social media applications that cater to the unique needs of older users. These tools will enable older adults to access health communication resources and bridge the gap of social media inequalities.

## Figures and Tables

**Table 1 healthcare-11-01455-t001:** Demographic and socioeconomic profile (*n* = 826).

Variable	*n*	%
**Age in years ^a^**		
Mean (SD)	67.6	6.4
**Gender**		
Male	400	48.4
Female	426	51.6
**Marital status**		
Single	32	3.9
Married	572	69.2
Divorced/Widowed/Single parent	222	26.9
**Highest qualification level**		
Primary	223	27.0
Secondary	291	35.2
Post-secondary education (Pre-university/Diploma)	140	16.9
Tertiary education (Degree/Master)	172	20.8
**Household income group (B40) ^b^**		
Low (B40)	530	64.2
Middle (M40)	221	26.8
High (T20)	75	9.1
**Income source**		
Work	201	24.3
Business/Investment	135	16.3
Children	287	34.7
Other sources	203	24.6
**Employment status**		
Employed	187	22.6
Not employed	285	34.5
Retired	354	42.9
**Living arrangement**		
Living alone at home	91	11.0
Living together with family	717	86.8
Living with others	18	2.2
**Residential area**		
Urban	581	70.3
Rural	245	29.7
**Health condition**		
With chronic illness	447	54.1
Without chronic illness	379	45.9

Notes: ^a^ presented as mean (SD). ^b^ B40 (<MYR 4850); M40 (MYR 4850–MYR 10,959); T20 (>MYR 10,960) [26].

**Table 2 healthcare-11-01455-t002:** Social media needs among older adults in Malaysia (*n* = 638).

Social Media Needs	*n* (%)	Mean (SD)
**Diversion (escapism and tension release)**		8.42 (1.88)
High (>6)	556 (87.1)	
Low (≤6)	82 (12.9)	
**Cognitive needs (information and knowledge acquisition)**		11.92 (2.37)
High (>8)	580 (90.9)	
Low (≤8)	58 (9.1)	
**Affective needs (emotions, pleasure, and feelings)**		7.06 (2.00)
High (>6)	347 (54.4)	
Low (≤6)	291 (45.6)	
**Personal integrative needs (enhancement of credibility and status)**		9.00 (2.73)
High (>8)	315 (49.4)	
Low (≤8)	323 (50.6)	
**Social integrative needs (interaction with friends and family)**		15.53 (2.74)
High (>10)	609 (95.5)	
Low (≤10)	29 (4.5)	

**Table 3 healthcare-11-01455-t003:** Quality of life among older adults in Malaysia (*n* = 826).

Quality of Life	n (%)	Mean (SD)
**Overall quality of life (OQOL)**		3.85 (0.78)
Very good (5)	135 (16.3)	
Good (4)	488 (59.1)	
Neither good nor poor (3)	164 (19.9)	
Poor (2)	26 (3.1)	
Very poor (1)	13 (1.6)	
**General health quality (GHQ)**		3.60 (0.88)
Very satisfied (5)	103 (12.5)	
Satisfied (4)	391 (47.3)	
Neither satisfied nor dissatisfied (3)	250 (30.3)	
Dissatisfied (2)	65 (7.9)	
Very dissatisfied (1)	17 (2.1)	
**Physical health quality (PHQ)**		62.55 (15.15)
Very good (76–100)	134 (16.2)	
Good (51–75)	487 (59.0)	
Medium (26–50)	134 (16.2)	
Poor (0–25)	11 (1.3)	
**Psychology health quality (PSYHQ)**		66.99 (13.33)
Very good (76–100)	163 (19.7)	
Good (51–75)	564 (68.3)	
Medium (26–50)	97 (11.7)	
Poor (0–25)	2 (0.2)	
**Social relationship quality (SRQ)**		67.09 (18.13)
Very good (76–100)	146 (17.7)	
Good (51–75)	473 (57.3)	
Medium (26–50)	181 (21.9)	
Poor (0–25)	26 (3.1)	
**Environmental health quality (EHQ)**		68.92 (13.40)
Very good (76–100)	184 (22.3)	
Good (51–75)	548 (66.3)	
Medium (26–50)	89 (10.8)	
Poor (0–25)	5 (0.6)	

**Table 4 healthcare-11-01455-t004:** Bivariate association between social media needs and quality of life (*n* = 638).

Variable	*n*	Physical Health			Psychological Health			Social Relationships			Environmental Health		
		Mean (SD)	*t*-Statistic	*p*-Value *	Mean (SD)	*t*-Statistic	*p*-Value *	Mean (SD)	*t*-Statistic	*p*-Value *	Mean (SD)	*t*-Statistic	*p*-Value *
**Diversionary needs**													
Low	82	61.49 (13.80)	−2.493	**0.013**	66.35 (12.91)	−1.500	0.134	68.83 (19.23)	0.374	0.708	69.76 (10.99)	−0.017	0.986
High	556	65.72 (14.41)			68.67 (13.10)			68.02 (18.08)			69.78 (13.89)		
**Cognitive needs**													
Low	58	59.53 (13.47)	−3.150	**0.002**	64.57 (13.09)	−2.330	**0.020**	67.36 (17.81)	−0.334	0.738	68.21 (11.15)	−0.927	0.354
High	580	65.74 (14.37)			68.76 (13.04)			68.20 (18.27)			69.94 (13.76)		
**Affective needs**													
Low	291	63.67 (14.39)	−2.419	**0.016**	68.62 (12.25)	0.428	0.669	67.99 (19.35)	−0.177	0.860	69.43 (13.58)	−0.599	0.550
High	347	66.43 (14.30)			68.17 (13.76)			68.24 (17.24)			70.07 (13.53)		
**Personal integrative needs**													
Low	323	63.70 (14.73)	−2.631	**0.009**	67.38 (12.41)	−2.330	0.052	66.99 (18.32)	−1.595	0.111	69.34 (13.24)	−0.830	0.407
High	315	66.69 (13.90)			69.40 (13.70)			69.29 (18.07)			70.23 (13.86)		
**Social integrative needs**													
Low	29	57.24 (14.41)	−3.056	**0.002**	61.55 (13.50)	−2.889	**0.004**	62.07 (19.62)	−1.836	0.067	63.76 (9.99)	−2.459	**0.014**
High	609	65.55 (14.30)			68.70 (12.99)			68.41 (18.11)			70.07 (13.63)		

Notes: significant values indicated in bold. * *t*-test.

**Table 5 healthcare-11-01455-t005:** Multiple linear regression for predicting quality of life.

Predictor Variable	Physical Health	Psychological Health	Social Relationships	Environmental Health
	*β* (CI)	*β* (CI)	*β* (CI)	*β* (CI)
**Age**	0.009 (−0.199; 0.246)	0.030 (−0.141; 0.285)	0.063 (−0.095; 0.517)	0.068 (−0.058; 0.4)
**Gender**				
Female	Ref	Ref	Ref	Ref
Male	−0.045 (−3.523; 0.93)	−0.086 (−4.39; −0.129) *	−0.044 (−4.656; 1.473)	−0.067 (−4.106; 0.484)
**Marital status**				
Single	Ref	Ref	Ref	Ref
Married	−0.110 (−8.906; 1.58)	−0.029 (−5.906; 4.127)	0.180 (0.388; 14.819) *	0.037 (−4.234; 6.574)
Divorced/Widow	−0.195 (−12.554; −1.44) *	−0.062 (−7.326; 3.307)	0.127 (−1.877; 13.417)	0.002 (−5.674; 5.781)
**Highest qualification**				
Primary	Ref	Ref	Ref	Ref
Secondary	−0.060 (−5.132; 1.493)	−0.145 (−7.153; −0.815) *	−0.069 (−7.197; 1.92)	−0.032 (−4.319; 2.509)
Post-secondary	−0.012 (−3.683; 2.853)	−0.107 (−6.631; −0.377) *	−0.088 (−8.485; 0.509)	−0.062 (−5.467; 1.269)
Tertiary	−0.098 (−7.895; 0.584)	−0.192 (−10.54; −2.427)**	−0.147 (−12.767; −1.098) *	−0.138 (−9.208; −0.469) *
**Household income group**				
Low (B40)	Ref	Ref	Ref	Ref
Middle (M40)	0.025 (−1.904; 3.472)	−0.035 (−3.551; 1.593)	0.004 (−3.561; 3.837)	0.027 (−1.985; 3.556)
High (T20)	0.027 (−0.991; 1.792)	0.023 (−1.017; 1.647)	−0.026 (−2.407; 1.424)	0.026 (−1.075; 1.794)
**Income source**				
Work	Ref	Ref	Ref	Ref
Business/Investment	0.041 (−1.259; 4.256)	0.087 (0.223; 5.499) *	0.030 (−2.401; 5.188)	0.012 (−2.43; 3.254)
Children	0.136 (1.276; 7.456) **	0.087 (−0.405; 5.508)	0.019 (−3.488; 5.017)	0.026 (−2.399; 3.971)
**Employment status**				
Not employed	Ref	Ref	Ref	Ref
Employed	0.162 (1.836; 8.695)	0.120 (0.249; 6.812) *	0.133 (0.742; 10.181) *	0.091 (−0.748; 6.321)
Retired	−0.009 (−3.052; 2.548)	0.061 (−1.066; 4.292)	0.020 (−3.109; 4.597)	0.024 (−2.221; 3.55)
**Living arrangement**				
Living alone	Ref	Ref	Ref	Ref
Living with family	0.029 (−2.249; 4.798)	0.066 (−0.733; 6.011)	0.025 (−3.481; 6.219)	0.039 (−2.026; 5.238)
Living with others	0.029 (−5.027; 11.218)	0.043 (−3.614; 11.931)	0.083 (0.001; 22.359)*	0.022 (−6.144; 10.6)
**Residential area**				
Urban	Ref	Ref	Ref	Ref
Rural	−0.008 (−2.708; 2.185)	0.006 (−2.178; 2.504)	0.016 (−2.707; 4.027)	−0.009 (−2.79; 2.254)
**Health condition**				
With chronic illness	Ref	Ref	Ref	Ref
Without chronic illness	0.322 (7.137; 11.371) ***	0.175 (2.559; 6.61) ***	0.114 (1.24; 7.067) **	0.102 (0.575; 4.938) *
**Social media needs**				
**Diversionary needs**	0.033 (−0.413; 0.923)	0.060 (−0.22; 1.058)	−0.039 (−1.295; 0.544)	−0.009 (−0.753; 0.623)
**Cognitive needs**	0.109 (0.116; 1.208) *	0.045 (−0.272; 0.773)	0 (−0.748; 0.755)	0.032 (−0.378; 0.748)
**Affective needs**	−0.067 (−1.197; 0.236)	−0.168 (−1.786; −0.414) **	−0.080 (−1.717; 0.256)	−0.056 (−1.121; 0.356)
**Personal integrative needs**	0.011 (−0.446; 0.567)	0.061 (−0.194; 0.776)	0.020 (−0.562; 0.833)	−0.010 (−0.571; 0.473)
**Social integrative needs**	0.120 (0.16; 1.105) **	0.174 (0.379; 1.284) ***	0.178 (0.533; 1.834) ***	0.156 (0.286; 1.26) **
	*R*^2^ = 0.223	*R*^2^ = 0.139	*R*^2^ = 0.080	*R*^2^ = 0.068
	Adjusted *R*^2^ = 0.195	Adjusted *R*^2^ = 0.108	Adjusted *R*^2^ = 0.047	Adjusted *R*^2^ = 0.034
	*F* ratio = 8.006,	*F* ratio = 4.520,	*F* ratio = 2.440,	*F* ratio = 2.028,
	*p* < 0.001	*p* < 0.001	*p* < 0.001	*p* < 0.01

Notes: *** *p* < 0.001; ** *p* < 0.01; * *p* < 0.05. *β* = standardized regression coefficient. CI = confidence interval.

## Data Availability

The data that support the findings of this study are available on request from the corresponding author. The data are not publicly available due to privacy or ethical restrictions.

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
