# Peer review of "Trends in Health Communication: Social Media Needs and Quality of Life among Older Adults in Malaysia"

_healthcare, 2023, doi:10.3390/healthcare11101455_

Round 1
Reviewer 1 Report
Dear authors,
This is a very well written manuscript.
Appropriate tools were used (WHOQOL-BREF and SNSUN scale), main results are presented as analytic text and through the understandable tables; discussion is correct and the conclusions are well drawn.
The only question is - why and how come that "a convenience sample" was investigated (instead of random one)?
Author Response
Dear reviewer,
We want to thank you for taking the time and effort to review the manuscript. We sincerely appreciate all your valuable comments, feedback and suggestions, which helped us improve the manuscript's quality. Attached are the documents for the revised manuscript and rebuttal report.

Reviewer 2 Report
Dear Authors,
Congratulations for your brilliant work.
However, one qualitative part could be enhanced your work further.
Could you explain more details about the data collection procedure and sample size calculation. More reference is required specially for the first part of your discussion.
All the best.
Author Response

(The authors gave the same response as above.)

Reviewer 3 Report
This is an interesting study conducted in Malaysia on social media needs and quality of life in older adults. It was overall well done and provided interesting results.
Abstract should be structured with labels e.g. background, methodology, results. Authors mentioned 826 participants twice in abstract. Methodology and results section in abstract can be rephrased to be more concise and clear. I feel the entire abstract needs rewriting as it is not clearly showcasing the findings in this study.
Main body is well written and organized.
Some limitations are mentioned in the conclusion, but a separate limitations section would be better.
Author Response

(The authors gave the same response as above.)

Reviewer 4 Report
The study addresses an important aspect of the digital divide for older people. The obtained results allow you to gain knowledge about the needs of social media. The theoretical part was prepared correctly. The methodological part is described in detail. It would be useful to indicate the objectives of the study or possibly research questions. Research tools are described in detail. The analysis of the results and the discussion were prepared as required. The authors indicated the limitations of their research (small research sample) and proposed areas for further exploration.
Author Response

(The authors gave the same response as above.)

Reviewer 5 Report
Dear Authors,
Thank you for the opportunity to review an interesting article entitled: 'Trends in health communication: social media needs and quality of life among older adults in Malaysia'. The aim of the article is to analyse the relationship of social media use and quality of life among Malaysian adults. The subject matter is interesting and extremely topical.
The strengths of the article presented for evaluation are the large sample size, the use of statistical analyses and the citation of current literature.
The reviewer's job, on the other hand, is to help improve the article so that it meets the highest possible standards of the journal, therefore I will focus on its weaknesses.
General comments
[1]. Text throughout the manuscript should be uniformly formatted.
[2]. The structure of the "Material and methods" section needs to be improved. The sections "Participants", "Design and procedure", "Data Collections Tools", "Data analysis" should be distinguished here.
[3]. The bibliography should be formatted according to the journal's guidelines.
Material and methods
[4]. The subjects should be described in the 'Participants' section (data from lines 190-196).
[5]. Do the SNSUN and WHOQOL-BREF scales have an adaptation in Malaysia? If yes, please provide a footnote. On the other hand, if not, please provide the adaptation process with the relevant psychometric coefficients.
[6]. What is Cronbach's alpha for the current study? Please complete.
Results
[7]. In line 182, the authors state that they used a t-test. However, nowhere in the analysis of results do they present the results of this test. Instead, only the p-value coefficient is given in Table 4. Providing only this coefficient does not entitle one to analyse differences between means.
[8]. In Tables 5 and 6 and their descriptions, apart from the β coefficient and its confidence intervals (it is not stated how they were calculated), there are no other regression coefficients: R, R2, cR2 and F. Only by completing this information will it be possible to fully evaluate the results presented and the conclusions drawn from them.
Author Response

(The authors gave the same response as above.)

Reviewer 6 Report
Thank you for the opportunity to review this interesting article. The article describes the results of a study about social networking needs and is based on a survey. The article is interesting and reflects a work that can be useful for developing initiatives based on these tools. However, some minor aspects could be revised to improve the article's readability. These aspects are the following:
In section 2.1., The authors should describe the type of study conducted, the eligible population, and the inclusion and exclusion criteria. These aspects are important because they will later allow an assessment of potential selection bias, which is one of the most important in this type of study.
In that same section, 2.1., I would remove the number of participants since this would be the first result of the study, given that it is obtained once the selection methods and the inclusion and exclusion criteria are applied to the eligible population.
I would also invite the authors to create a specific section describing the ethical aspects in detail instead of being described in this section which, according to the title, describes other aspects.
In section 2.2. the authors point out that they use a questionnaire that they have translated. Given that the questionnaire is the tool on which their research is structured and the article that describes it, it is essential that they provide much more detail: for example, whether the questionnaire is original, adapted, or from other authors. Where possible, add citations and references to sources. Another important aspect is to detail the translation process, as it may or may not have involved cross-cultural adaptation. This is important because translation and cross-cultural adaptation also have an associated bias. It would also be helpful for the authors to describe whether the questionnaire required validation and, if so, how this was done (e.g., by exploratory factor analysis, confirmatory factor analysis, or both, and what other analyses confirmed the validity and consistency of the questionnaire).
In section 3.1, the first result to be shown should be the number of participants obtained.
The authors should include a section on limitations and strengths at the end of the discussion. Curiously, they cite some limitations at the end of the conclusions when I think that would fit much better at the end of the discussion. Some of the important limitations in this type of study are selection bias, the validity and reproducibility of the tool used, and an adequate assessment of the external validity of the results and conclusions obtained.
Author Response

(The authors gave the same response as above.)

Round 2
Reviewer 5 Report
Dear Authors,
Most of the comments were taken into account, however:
[1]. The 'Participants' section should characterise the people to be surveyed (this description is currently in section '3.1. Demographic and socioeconomic profile'), rather than simply discussing how to count the size of the required population to be surveyed.
[2]. The notation of the designations 'p', 'r', 't', 'R2', 'β', etc., which should be italicised in each case, should be standardised.
Author Response
Dear reviewer,
We want to thank you for taking the time and effort to review the manuscript. We sincerely appreciate all your valuable comments, feedback and suggestions, which helped us improve the manuscript's quality. Attached are the documents for the second round revised manuscript and rebuttal report.
